# Based on the Feedforward Inputs Obtained by the Intelligent Algorithm the Moving Mirror Control System of the Fourier Transform Spectrometer

Ying Huang [1,2,3], Juan Duan [1,2,*], Qian Guo [1,2], Zhanhu Wang [1,2,*] and Jianwen Hua [1,2]

1   Shanghai Institutes of Technical Physics, Chinese Academy of Sciences, Shanghai 200083, China; huangying@mail.sitp.ac.cn (Y.H.); guoqian@mail.sitp.ac.cn (Q.G.); jwhua@mail.sitp.ac.cn (J.H.)
2   Key Laboratory of Infrared System Detection and Imaging Technology, Chinese Academy of Sciences, Shanghai 200083, China
3   University of Chinese Academy of Sciences, Beijing 100049, China
*   Correspondence: duanjuan@mail.sitp.ac.cn (J.D.); wangzhanhu@mail.sitp.ac.cn (Z.W.)

**Abstract:** A moving mirror control system of the Fourier transform spectrometer (FTS) based on the feedforward inputs obtained by the intelligent algorithm is proposed in this paper. Feedforward control is an important part of the moving mirror speed control system of the FTS. And it is always difficult to quantitatively calculate the feedforward inputs through a precise mathematical model of the controlled object. Therefore, based on the expected motion law, an intelligent adaptive algorithm for obtaining feedforward inputs of the moving mirror system was designed. The algorithm decomposed the motion stroke into several position points, iteratively obtained the driving quantity of the moving mirror that met the expected instantaneous speed of each position point, and finally obtained the feedforward inputs of the whole motion stroke. The feedforward inputs obtained by the intelligent algorithm combined with the speed loop PID control constitute the complete moving mirror speed control system. Then, we applied the control system to the moving mirror of the FTS and acquired the velocity of the moving mirror. The experimental results show that the control system is feasible, the error of the peak-to-peak velocity is 0.047, and the error of the root mean square (RMS) velocity is 0.003. Compared with the single-speed-loop control system without feedforward inputs, the error of the peak-to-peak velocity is reduced by 43.3%, and the error of the RMS velocity is reduced by 67.7%, realizing a more accurate control of the moving mirror. Therefore, the control system based on the feedforward inputs obtained by the intelligent algorithm is a feasible and effective moving mirror speed control scheme of the FTS.

**Keywords:** fourier transform spectrometer; linear motor; feedforward control; intelligent algorithm; velocity uniformity



## 1. Introduction

The Fourier transform spectrometer (FTS) is an important spaceborne infrared remote sensing instrument. It obtains the spectral information of atmospheric radiation by collecting and Fourier-transforming the interference signal of atmospheric infrared radiation, and then obtains the vertical distribution of atmospheric gas composition, humidity, and temperature [1–5]. The Chinese polar-orbiting meteorological satellite series FY-3 and geostationary meteorological satellite series FY-4 [6,7], the American National Polar-orbiting Operational Environmental Satellite System (NPOESS) [8–11] and the Japanese greenhouse gas observation satellite (GOSAT) [12,13] all carry this important payload. In order to detect the radiation of different spectra, spectrometers need to split the spectrum of atmospheric radiation, which means spreading the spectral radiance according to the wavelength or frequency of light. There are three main ways of splitting: splitting by filter, splitting by grating, and splitting by interference. The FTS in this paper uses the same way of splitting

as the classical Michelson interferometer [14–17], and the moving mirror is one of the core components. The uniformity and stability of the moving mirror's velocity directly affect the quality of the interferogram, so it must have high-precision control. The elastic translational supporting mechanism is widely used in the motion system of the moving mirror of spaceborne FTS, which has the remarkable characteristics of being frictionless and lubrication-free, as well as having a long life [18–20]. The driving structure of the moving mirror of the FTS used in this paper adopts the elastic translational supporting mechanism and linear motor, as shown in Figure 1. The moving mirror is driven by a linear motor and makes a reciprocating linear motion. Therefore, the control of the moving mirror can be transformed into the control of the linear motor.

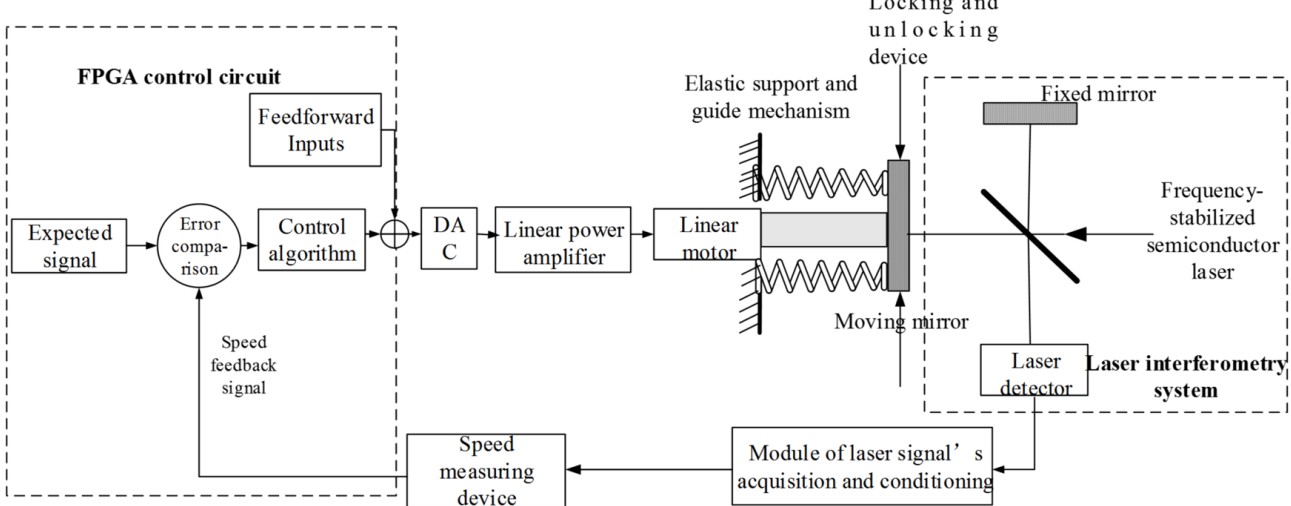

**Figure 1.** The moving mirror speed control system of the FTS.

For servo-motor control systems, feedback control is commonly used, and is the main means of control [21–25].

A moving mirror control system for spaceborne FTS is described in the literature [26]. Its motion system consists of elastic supports, flat mirrors, and a voice coil motor. The control system adopts single position-loop control to achieve control of the position and speed of the moving mirror. The final control effect is that the error of the peak-to-peak velocity in the uniform speed zone is 0.088, and the error of the RMS velocity is 0.014. The Miniature Thermal Emission Spectrometer (Mini-TES) is a single-pixel FTS used to measure thermal emission for mapping surface minerals on Mars. The control system of the moving mirror is a speed feedback controller, which uses a voice coil motor. The servo is a digital servo that counts the time between a fringe signal's zero crossings to generate a velocity error signal. The analog portion of the servo uses the tachometer signal from the motor to assist in hedback compensation [27]. The Atmospheric Chemistry Experiment (ACE) is the mission selected by the Canadian Space Agency for its next science satellite, SCISAT-1. The infrared FTS is the primary element. In the control system, the servo motor provides position control of the rotary arm in a closed-loop configuration. The position profile is generated by an FPGA. A Proportional-Integrator-Derivative-Filter (PIDF) servo compensation is implemented and coupled with a feedforward controller [28]. In [29,30], the authors introduce feedforward control into the linear motor servo system, which shortens the adjustment time of the system and reduces the overshoot of the system to achieve higher precision control.

Once the feedforward control is introduced into the control system, the response of the moving mirror to the desired command can be greatly improved. However, for the moving mirror speed control system of the FTS, to obtain accurate feedforward inputs, the controlled object and the power converter need to be modeled very accurately, which is

quite difficult. Therefore, the moving mirror control system with feedforward control is rarely applied.

In this paper, an intelligent algorithm that can obtain the feedforward inputs automatically is proposed. It is very convenient to obtain accurate feedforward inputs using this algorithm. The obtained feedforward inputs and the feedback control quantity were added to jointly control the moving mirror movement. This control scheme was applied to the moving mirror system of the FTS, driving the moving mirror's reciprocating motion, and the speed of the moving mirror is collected at the same time. With the feedforward inputs of the intelligent algorithm, the error of the peak-to-peak velocity is reduced by 43.3%, and the error of the RMS velocity is reduced by 67.7%. The experimental results show that the moving mirror control system based on the feedforward inputs obtained by the intelligent algorithm can cause the moving mirror to achieve better velocity uniformity than the single-speed-loop control system without feedforward inputs, realizing more accurate control.

The control system proposed in this paper, based on the feedforward inputs obtained by the intelligent algorithm, is a feasible and effective moving mirror speed control scheme of the FTS.

## 2. Materials and Methods

The moving mirror speed control system of the FTS presented in this paper is mainly composed of a speed feedforward controller, speed feedback controller, actuator, and linear motor [31,32], as shown in Figure 2. In the figure, $\beta$ is the magnification of the actuator, $L$ is the inductance of the coil, $R$ is the resistance of the coil, $K_f$ is the force constant of the motor, $m$ is the mass of the motor, $K_x$ is the axial stiffness of the spring, $K_{be}$ is the back electromotive force coefficient, $U_q(i)$ is the feedforward input, $U_f(i)$ is the feedback control quantity, and $U(i)$ is the total control quantity. Since $U(i) = U_q(i) + U_f(i)$, the key to this system is finding $U_q(i)$ and $U_f(i)$, where $U_q(i)$ is obtained by the intelligent algorithm, and $U_f(i)$ is calculated by the feedback control loop.

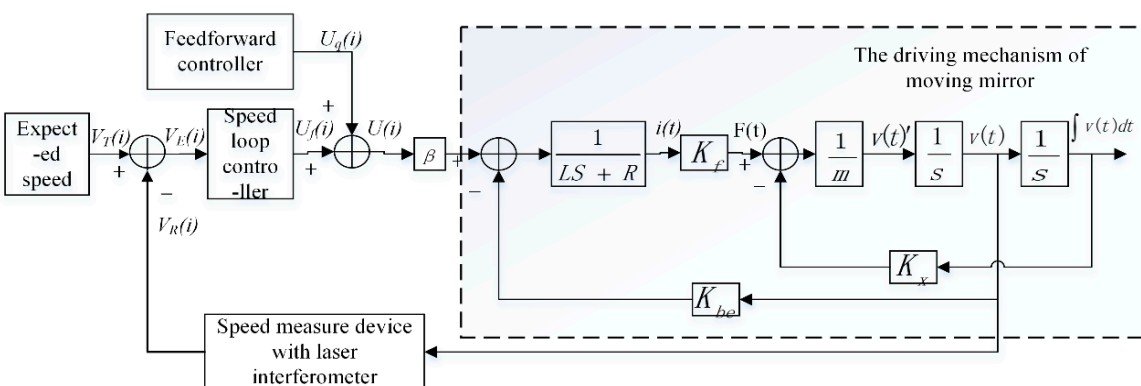

**Figure 2.** The block diagram of the moving mirror speed control system.

In practical applications, because a mathematical model of the actuator and motor system cannot be accurately obtained, it is very difficult to obtain accurate feedforward inputs through theoretical calculation. We urgently need to study an automatic intelligent algorithm that can find the feedforward inputs.

### 2.1. Intelligent Algorithm for Finding Feedforward Inputs

The intelligent algorithm is a kind of speed feedforward algorithm; it acquires the feedforward inputs according to the expected speed law. In this paper, the expected velocity motion law of the moving mirror consists of three parts: the accelerating section, uniform

section, and decelerating section. The relationship between the velocity $v$ and the time $t$ is as follows:

$$v(t) = \begin{cases} v_m \cdot \sin\left(\dfrac{\pi}{2} \cdot \dfrac{t}{t_1}\right) & 0 \le t < t_1 \\ v_m & t_1 \le t < T - t_1 \\ v_m \cdot \sin\left(\dfrac{\pi}{2} \cdot \dfrac{T-t}{t_1}\right) & T - t_1 \le t \le T \end{cases} \quad , \tag{1}$$

where $v_m$ is the velocity of the uniform section, $T$ is the single-pass time of movement, and $t_1$ is the time of the accelerating section.

The core idea of the intelligent algorithm is to divide the whole motion stroke of the moving mirror into several position points according to a certain rule, and obtain the driving voltage as the feedforward input of each position point, which will cause the speed of each point in the movement process to meet the expected speed range.

2.1.1. Design of Set Values in Algorithm

The moving mirror system of the FTS in this paper has a reference laser interference system, as shown in Figure 1, where the semiconductor laser is used as the reference light source, and the silicon photodetector is used to complete the photoelectric conversion of the laser signal. Then, the periodic square wave signal is obtained by filtering and shaping the converted electrical signal, which is the exact laser interference signal. When the laser wavelength is $\lambda$, every $\dfrac{\lambda}{2}$ movement of the moving mirror generates one laser interference signal, so the signal can be used to determine the direction of movement, position, and speed of the moving mirror. In a digital processor, we usually measure how much $\dfrac{\lambda}{2}$ the moving mirror displacement contains to represent the digital displacement.

The key point of the algorithm design is to find the set values $S(i)$, $V_{TL}(i)$, and $V_{TH}(i)$. $S(i)$ is the digital displacement at the time $i \times t_0$, while $t_0$ is the control period. $V_{TL}(i)$ and $V_{TH}(i)$ are the upper limit and the lower limit of the expected speed at $S(i)$, respectively. The train of thought is shown in Figure 3. First, divide the motion period into $\dfrac{T}{t_0}$ segments, and find the displacement $S(i)$ ($0 \le i \le \dfrac{T}{t_0}$). Since this algorithm is created according to the instantaneous velocity at $S(i)$, it is necessary to find the exact time point $t(S(i))$ corresponding to $S(i)$ first, and then that according to Equation (1), the instantaneous velocity $v(t(S(i)))$ can be obtained. Finally, according to the $T$ method, the upper limit of the expected speed $V_{TL}(i)$ and the lower limit of the expected speed $V_{TH}(i)$ are obtained.

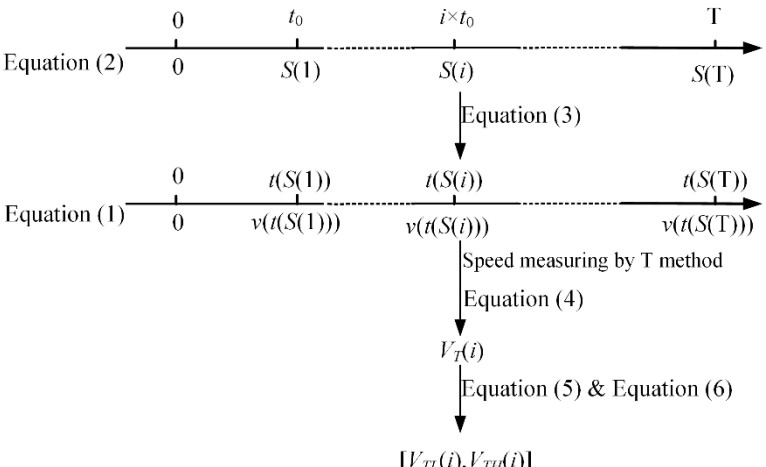

**Figure 3.** The train of thought to find the set values $S(i)$, $V_{TL}(i)$, and $V_{TH}(i)$.

According to the requirements of the reciprocating uniform linear motion of the moving mirror system, the single-pass expected velocity law is first established as shown in Equation (1), before it is integrated, divided by $\frac{\lambda}{2}$, and rounded; then, the single-pass expected displacement law is obtained, as shown in Equation (2).

$$S(t) = \begin{cases} \dfrac{-2t_1}{\pi} \cdot v_m \cdot \left( \cos\left( \dfrac{\pi}{2} \cdot \dfrac{t}{t_1} \right) - 1 \right) & 0 \leq t < t_1 \\[3mm] v_m \cdot (t - t_1) + \dfrac{2t_1}{\pi} \cdot v_m & t_1 \leq t < T - t_1, \\[3mm] v_m \cdot (T - 2t_1) + \dfrac{2t_1}{\pi} \cdot v_m \cdot \cos\left( \dfrac{\pi}{2} \cdot \dfrac{T - t}{t_1} \right) & T - t_1 \leq t \leq T \end{cases} \qquad (2)$$

The control period of the system in this paper is $t_0$ ms, substituting $i \times t_0$ into Equation (2) to obtain the displacement $S(i)(0 \leq i \leq \frac{T}{t_0})$.

How to find the inverse function of $S(t)$, and the expression of time $t$ with respect to displacement $S$ is shown in Equation (3);

$$t(S) = \begin{cases} \dfrac{2t_1}{\pi} \cdot \cos^{-1}\left[ \left( \lambda \cdot S - \dfrac{2t_1 \cdot v_m}{\pi} \right) \left( \dfrac{-\pi}{2t_1 \cdot v_m} \right) \right] & 0 \leq S < \dfrac{2t_1 \cdot v_m}{\pi \cdot \lambda} \\[3mm] \dfrac{\lambda \cdot S}{v_m} - \dfrac{2t_1}{\pi} + t_1 & \dfrac{2t_1 \cdot v_m}{\pi \cdot \lambda} \leq S < \dfrac{v_m[2t_1 + \pi(T - 2t_1)]}{\pi \cdot \lambda} \\[3mm] T - \dfrac{2t_1}{\pi} \cdot \cos^{-1}\left[ \dfrac{\pi}{2t_1} \left( \dfrac{\lambda \cdot S}{v_m} - \dfrac{2t_1}{\pi} - T + 2t_1 \right) \right] & \dfrac{v_m[2t_1 + \pi(T - 2t_1)]}{\pi \cdot \lambda} \leq S \leq \dfrac{v_m[4t_1 + \pi(T - 2t_1)]}{\pi \cdot \lambda} \end{cases} \qquad (3)$$

By substituting each $S(i)$ into Equation (3), the time $t(S(i))$ corresponding to $S(i)$ can be obtained. Then, by substituting $t(S(i))$ into Equation (1), we can finally obtain the expected instantaneous velocity $v(t(S(i)))$ of the moving mirror at $S(i)$.

The traditional method of measuring the moving mirror's velocity by reference laser interference signal measures the number of laser interference signals in a period, which is easy to implement but has low measurement accuracy. So, in order to improve the accuracy and use the digital quantity of the computer to represent these instantaneous velocity values accurately, we adopted the T speed measuring method [33]. The method measures the time of the moving mirror over a distance by counting the number of clock pulses. Since the frequency of the clock pulse can be very high, the measuring accuracy greatly improves. The frequency of the clock pulses of this algorithm was 1 MHz. In a digital processor, by using the number of clock pulses within the $\frac{\lambda}{2}$ before $S(i)$ as the approximate representation of the instantaneous velocity value at $S(i)$, we can obtain the instantaneous velocity $V_T(i)$ expressed by digital quantity, as shown in Equation (4).

$$V_T(i) = \frac{\dfrac{\lambda/2}{v(t(s(i)))}}{1/1 \times 10^6}, \qquad (4)$$

Since the displacement $\frac{\lambda}{2}$ is a constant number, the time and the speed are inversely proportional. In the real system, as that the speed is not an ideal constant value, we need to provide a reasonable speed error range. The smaller the range, the more accurate the results, and the higher the requirements for the anti-interference ability of the system. The algorithm running time will also be greatly increased. Therefore, we need to find a compromise between these aspects. Assuming that the allowable error range of $V_T(i)$ is $\pm 10\%$, the upper speed limit and the lower limit of the number of clock pulses is shown in Equation (5); the lower speed limit and the upper limit of the number of clock pulses is shown in Equation (6):

$$V_{TL}(i) = \frac{\dfrac{\lambda/2}{1.1v(t(s(i)))}}{1/1 \times 10^6},$$ (5)

$$V_{TH}(i) = \frac{\dfrac{\lambda/2}{0.9v(t(s(i)))}}{1/1 \times 10^6},$$ (6)

After the above steps, the set values $S(i)$, $V_{TL}(i)$, and $V_{TH}(i)$ in the algorithm are obtained.

2.1.2. Design of the Algorithm

The idea of the algorithm is to find the feedforward voltage $U_q(i)$ that satisfies the desired velocity range $[V_{TL}(i), V_{TH}(i)]$ at each position point $S(i)$ in order. In each complete search process, the moving mirror starts from a fixed point. When reaching $S(i)$, the $V_T(i)$ of the moving mirror is judged. If the $V_T(i)$ is in the range $[V_{TL}(i), V_{TH}(i)]$, the current driving voltage is recorded as $U_q(i)$ and the moving mirror returns to the starting point to prepare for finding the next feedforward voltage $U_q(i + 1)$; if the $V_T(i)$ is not in the range, the driving voltage is adjusted according to the $V_T(i)$, and the moving mirror also returns to the starting point to prepare to find $U_q(i)$.

The following variables need to be defined in the algorithm:

1.  *memory*, the array that is used to store the obtained feedforward inputs $U_q(i)s$ that meet the speed requirements, with a total of $\dfrac{T}{t_0}$ values being stored;

2.  $\Delta U$, the adjustment of driving voltage;

3.  *fh_flag*, the flag of the movement process. Since the search algorithm is only executed in the single-pass movement, the motion direction of the moving mirror in the search process is set as positive, *fh_flag* = 0, and the return process is set as negative, *fh_flag* = 1;

4.  *QL*, the same as *the i* of $U_q(i)$. If the $V_T(i)$ satisfies the expected range in the current search process, *QL* will be increased by 1; otherwise, *QL* will remain unchanged and its value range is $[1, \dfrac{T}{t_0}]$;

5.  *QM*, the variable of position point $S(QM)$, which is moving in the current search process. After determining the relationship between *QM* and *QL*, the driving voltage *memory(QM)* or the initial value *U0* of $Uq(i)$ are output. When *realS* = $S(QM)$, *QM* plus 1 is used until *QM* = *QL* and the *realS* reaches the last $S(QM)$; that is, $S(QL)$, after checking whether the speed is in the range $[V_{TL}(QM), V_{TH}(QM)]$. Then, *U0* is adjusted based on the check result. The value range of *QM* is $[0, QL]$;

6.  *realS*, the current real-time digital displacement of the moving mirror;

7.  *realV*, the real-time speed $V_T(QM)$ of the current position point $S(QM)$.

As shown in Figure 4, this is the complete search process of the moving mirror's forward motion when the return flag *fh_flag* = 0. The moving mirror starts from a fixed starting position (mechanical limit):

1.  When $0 \le QM \le QL - 2$, at *realS* = $S(QM)$, the driving voltage of moving mirror $U$ = *memory(QM)*;

2.  When *QM* = *QL* − 1, at *realS* = $S(QM)$, the driving voltage of the moving mirror $U$ = *U0*, and *U0* is the initial value of feedforward voltage at the current position;

3.  When *QM* = *QL*, *realV* is compared with $V_{TL}(QM)$ and $V_{TH}(QM)$ at *realS* = $S(QM)$. If $V_{TL}(QM) \le realV \le V_{TH}(QM)$, this indicates that *U0* meets the requirements. *U0* is stored in *memory* and *QL* is added individually, which means the feedforward voltage at the current position point is found. To find the feedforward voltage of the next position point, *fh_flag* is set to 1. If *realV* < $V_{TL}(QM)$, this indicates that the driving voltage is too big; then, $U$ = *U0* − $\Delta U$ and *fh_flag* is set to 1. If *realV* > $V_{TH}(QM)$, this indicates that the driving voltage is too small; then, $U$ = *U0* + $\Delta U$ and *fh_flag* is set to 1.

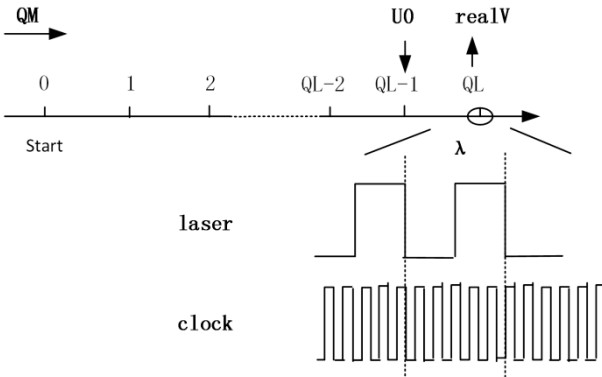

**Figure 4.** The schematic diagram of T-method velocity measurement of the moving mirror's forward motion in the search process.

When the return flag *fh_flag* = 1, the driving voltage of the moving mirror increases linearly so that the moving mirror slowly returns to the starting position. After the moving mirror arrives at the starting position, *fh_flag* is reset to 0, and the moving mirror starts from the starting point again, repeating the above steps until the feedforward voltages *Uq(i)* of $\frac{T}{t_0}$ position points *S(i)* are obtained. The flow chart of the algorithm is shown in Figure 5.

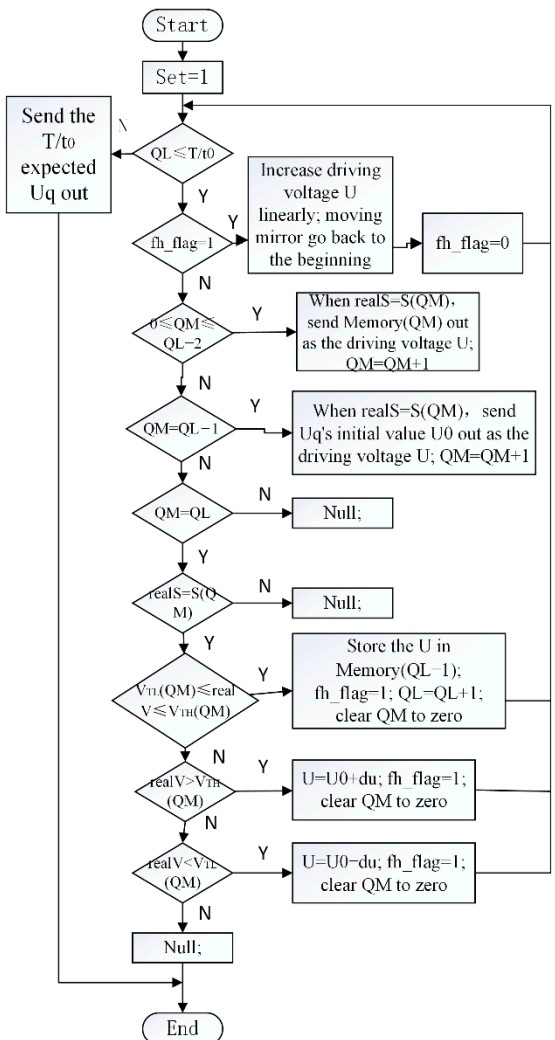

**Figure 5.** The algorithm flow chart for the intelligent acquisition of feedforward inputs.

In this paper, based on the Actel A3PE3000 hardware platform, the software program of the above algorithm was written and implemented in the Libero v9.1 development environment. After running this in the actual moving mirror system, the feedforward inputs $U_q(i)$ were obtained.

The advantage of this algorithm is that the moving mirror can obtain the feedforward voltage $U_q(i)$ intelligently in the real system, which greatly reduces the calculation workload. At the same time, because the running results are more consistent with the actual mechanical and electrical characteristics of the system, the feedforward inputs that are obtained are more accurate and effective. If the desired motion law changes, $S(i)$, $V_{TL}(i)$, and $V_{TH}(i)$ can be replaced in the program with new values, and then the program can be run again to search for the new feedforward voltage $U_q(i)$. If the parts (such as the linear motor) or the driving structure of the moving mirror motion system change, causing the mechanical and electrical characteristics of the system to change too, we only need to run the program with the new system again to obtain the new feedforward inputs that meet the speed requirement, which is very simple and convenient.

### 2.2. Finding of Feedback Control Quantity

The moving mirror control period of the interferometer presented in this paper is $t_0$ ms. Through the reference laser interference system, the speed of the moving mirror is measured by the T method high-frequency sampling, that is, the number of clock pulses $V_R(i)$ in the first $\frac{t_0}{2}$ of the $i$th control period are measured, and then subtracted from the expected number of clock pulses $V_T(i)$ to obtain the speed error $V_E(i)$. In the second $\frac{t_0}{2}$ of the $i$th control period, the error $V_E(i)$ is regulated to obtain the feedback control quantity $U_f(i)$ of the $i$th period.

The PID control algorithm is the most classic and commonly used feedback control algorithm, which can correct errors in the control system and discretize them to obtain the digital expressions, as shown in Equation (7):

$$U_f(i) = K_P \cdot V_E(i) + T_I\sum_{j=0}^{i} V_E(j) + T_D[V_E(i) - V_E(i-1)], \tag{7}$$

where $K_P$, $T_I$ and $T_D$ are proportional, integral, and differential parameters, respectively; $V_E(i)$ is the speed error at $S(i)$; $V_E(i-1)$ is the speed error at $S(i-1)$.

The integral term in Equation (7) needs to accommodate the errors of the past, and when performing algorithmic processing in the FPGA, this needs to occupy more bit length, resulting in the initial bit length of $V_E(i)$ being longer. The bit length is further lengthened after multiplication, which eventually leads to the bit length of $U_f(i)$ being too long. Due to the limitations regarding the number of bits of the DAC, in order to obtain good control accuracy, it is necessary to truncate the control quantity $U_f(i)$, which is complex, and the initial bit length of the accumulation is not easy to determine. Thus, a slight transformation of Equation (7) yields Equation (8):

$$U_f(i-1) = K_P \cdot V_E(i-1) + T_I\sum_{j=0}^{i-1} V_E(j) + T_D[V_E(i-1) - V_E(i-2)], \tag{8}$$

Equations (7) and (8) can be subtracted to obtain the incremental PID control expression, as shown in Equation (9):

$$U_f(i) = U_f(i-1) + (K_P + T_I + T_D)V_E(i) + (-K_P - 2T_D)V_E(i-1) + T_D \cdot V_E(i-2), \tag{9}$$

where the initial values of $U_f(i-1)$, $V_E(i)$, $V_E(i-1)$, and $V_E(i-2)$ are all 0. In this way, problems such as excessive bit length and complicated truncation can be effectively avoided. Using Equation (9), we find the feedback control quantity $U_f(i)$.

The feedforward inputs $U_q(i)$ obtained by the intelligent algorithm corresponding to each time $i \times t_0$ are stored in the control program in the form of an array. At time $i \times t_0$, the

$U_q(i)$ and the $U_f(i)$ are added to obtain the total control quantity $U(i)$, which is output to the driver finally.

## 3. Results

The experimental device of the system is mainly composed of the interferometer body, the control circuit, the power amplifier circuit and the reference laser interference system. The moving mirror control program is written to the FPGA, and the digital control quantity output from the FPGA is converted into an analog voltage by the DAC, and then applied to the linear motor after linear amplification. In the closed-loop mode, the speed of the moving mirror is collected, and the speed of the moving mirror under single-speed-loop control without feedforward inputs is collected for comparison.

Using a data acquisition card, the laser interference signal is acquired at high speed with a 2 MHz clock, as shown in Figure 6.

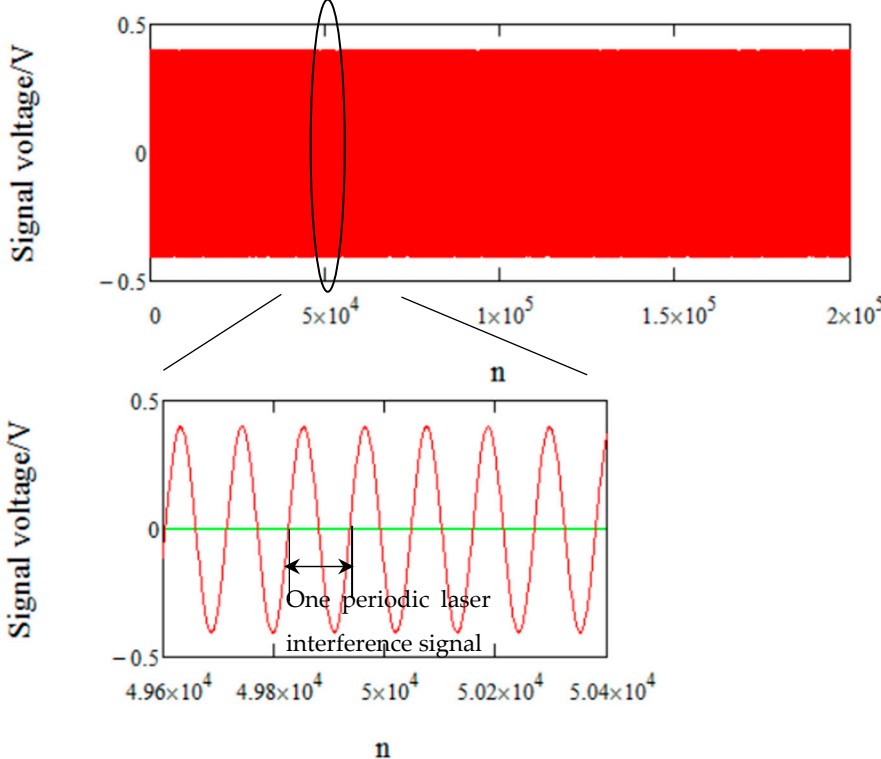

**Figure 6.** Laser interference signal sampled with a 2 MHz clock and its partial enlarged image.

The time that the moving mirror moves $\frac{\lambda}{2}$ using the number of sampling points $n_k$ in the $k_{\text{th}}$ laser interference signal for calculation is $\frac{n_k}{2 \cdot 10^6}$, and then is divided $\frac{\lambda}{2}$ by the time to obtain the instantaneous speed $V_k$ of the $k_{\text{th}}$ laser interference signal of the moving mirror, that is, $V_k = \dfrac{\frac{\lambda}{2}}{\frac{n_k}{2 \cdot 10^6}}$, so as to obtain the velocity curve of the whole motion stroke. In this paper, we are mainly concerned with the velocity's uniformity in the uniform speed zone. Since the stroke in the uniform speed zone is 8.4 mm, $\lambda$ is 852.3 nm, $\dfrac{8.4}{\frac{\lambda}{2}} = 19{,}712$, we can intercept 9903 laser interference signals from the midpoint to each side, which can completely cover the uniform speed zone. The intercepted velocity curve of the uniform speed zone is shown in Figure 7 (red curve), and compared with the expected velocity curve (blue curve).

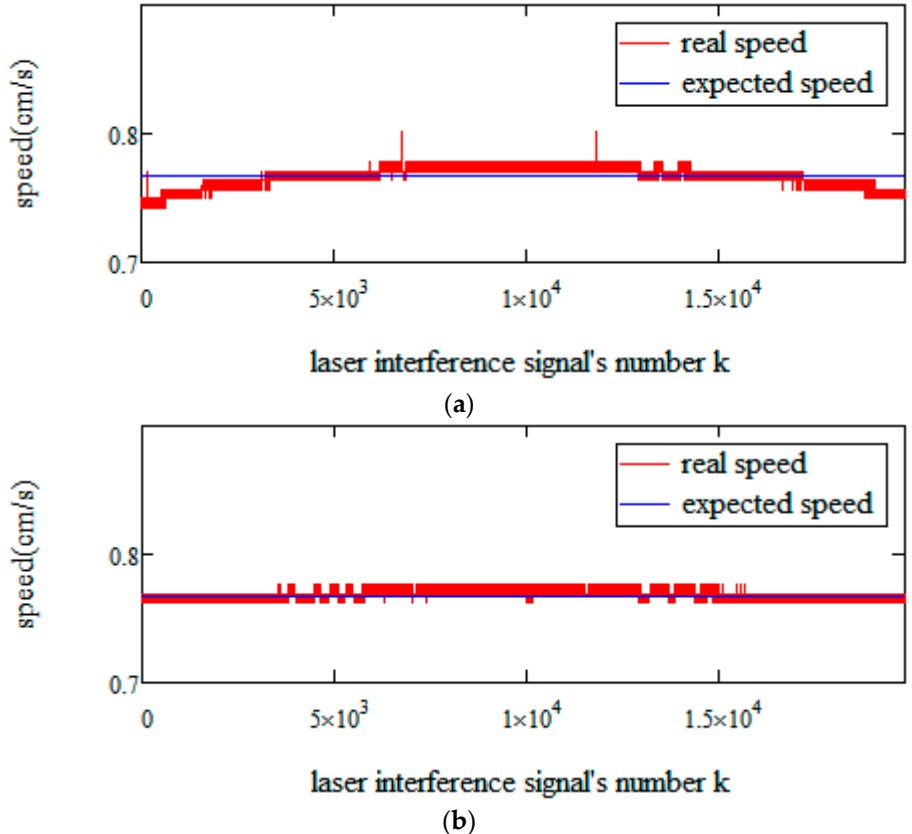

**Figure 7.** (**a**) Velocity curve of moving mirror without feedforward inputs in single-speed-loop control compared with the expected speed; (**b**) velocity curve of moving mirror using feedforward inputs obtained from intelligent algorithm in speed-loop control compared with the expected speed.

There are two main indicators tht can describe the uniformity of the moving mirror velocity: the error of the peak-to-peak velocity $V_{PP}$ and the error of the RMS velocity $V_{RMS}$, which can be calculated by Equations (10) and (11), respectively. We randomly selected 10 pieces of one-way velocity data and calculated the $V_{PP}$ and $V_{RMS}$ for each one-way datum, as shown in Tables 1 and 2. The obtained results were plotted as shown in Figures 8 and 9. They indicate that the moving mirror velocity has long-term stability.

$$V_{PP} = \frac{V_{MAX} - V_{MIN}}{V_A}, \tag{10}$$

$$V_{RMS} = \frac{\sqrt{\frac{\sum_0^{19,805}(V_k - V_A)^2}{19,806}}}{V_A}, \tag{11}$$

where $V_A$ is the average value of the velocity; $V_{MAX}$ and $V_{MIN}$ are the maximum and minimum values of the velocity.

**Table 1.** Ten sets of $V_{PP}$.

| With/Without Feedforward Inputs | 1 | 2 | 3 | 4 | 5 | 6 | 7 | 8 | 9 | 10 |
|---|---|---|---|---|---|---|---|---|---|---|
| Without | 0.085 | 0.083 | 0.088 | 0.083 | 0.083 | 0.083 | 0.087 | 0.083 | 0.083 | 0.087 |
| With | 0.047 | 0.047 | 0.043 | 0.046 | 0.047 | 0.047 | 0.047 | 0.043 | 0.047 | 0.047 |

**Table 2.** Ten sets of $V_{RMS}$.

| With/Without Feedforward Inputs | 1 | 2 | 3 | 4 | 5 | 6 | 7 | 8 | 9 | 10 |
|---|---|---|---|---|---|---|---|---|---|---|
| Without | 0.0094 | 0.0093 | 0.0097 | 0.0093 | 0.0093 | 0.0093 | 0.0096 | 0.0093 | 0.0093 | 0.0096 |
| With | 0.0030 | 0.0030 | 0.0029 | 0.0030 | 0.0030 | 0.0030 | 0.0030 | 0.0029 | 0.0030 | 0.0030 |

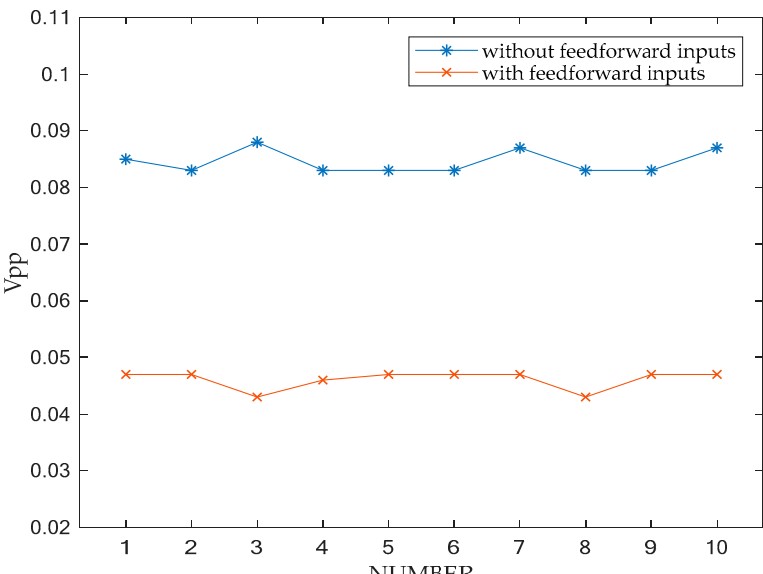

**Figure 8.** The $V_{PP}$ of each controller.

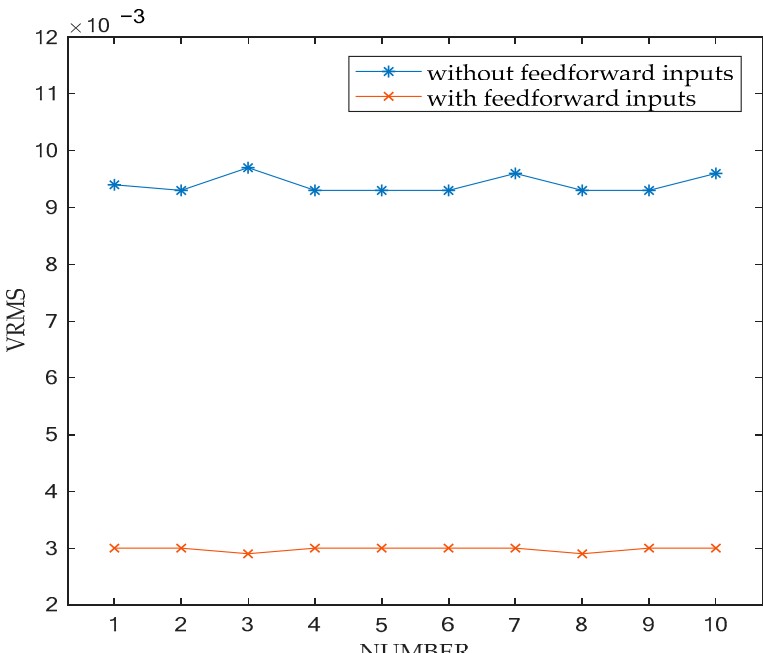

**Figure 9.** The $V_{RMS}$ of each controller.

It can be seen from the statistical results that, among the 10 sets of data, when the moving mirror control system based on the feedforward inputs of the intelligent algorithm is adopted, the maximum of the $V_{PP}$ is 0.047, and the maximum of the $V_{RMS}$ is 0.0030. When using the moving mirror control system without feedforward inputs, the minimum of the $V_{PP}$ is 0.083, and the minimum of the $V_{RMS}$ is 0.0093. The improvement in the control system's performance can be defined as the difference between the minimum velocity error

of the party with a poor control effect minus the maximum velocity error of the party with a better control effect. From Equations (12) and (13), it can be seen that, compared with the case where no feedforward inputs were used, the moving mirror control system based on the feedforward inputs obtained by the intelligent algorithm reduces the $V_{PP}$ by 43.4% and the $V_{RMS}$ by 67.7%.

$$\frac{0.083 - 0.047}{0.083} \times 100\% = 43.4\%, \tag{12}$$

$$\frac{0.0093 - 0.0030}{0.0093} \times 100\% = 67.7\%, \tag{13}$$

In addition to the uniformity of velocity, the sampling error and accuracy are also important indicators to evaluate the control system. In this paper, the moving mirror control system adopts the T method to measure speed. The laser interference signal is collected by a 2 MHz clock to measure the speed of the moving mirror. The control period $t_0$ is 1 ms, so the sampling error of the system can be obtained by Equation (14). The accuracy is defined as the ratio of the difference between the expected speed and the actual average speed $V_A$ and the desired speed, as shown in Equation (15), where the expected speed is 0.767 cm/s, the $V_A$ of the control system with the feedforward inputs of the intelligent algorithm is 0.7668 cm/s, and the $V_A$ of the control system without feedforward inputs is 0.7663 cm/s.

$$SE = \frac{1}{\frac{t_0}{2} * 2 \times 10^6}, \tag{14}$$

$$Ac = \frac{0.767 - V_A}{0.767}, \tag{15}$$

In the literature [26], the moving mirror control system is a single-position-loop control system without feedforward inputs. The speed measurement adopts the M method, that is, counting the number of laser interference signals in a control period (the number should be 18 in the literature [26]), and its sampling error is $1/18 \approx 0.056$. The accuracy is 0.0013. The $V_{PP}$ is 0.088, and the $V_{RMS}$ is 0.0140. In the literature [33], the sampling error of the control system is $1/40,000 = 0.000025$, and the $V_{RMS}$ is 0.0102. A comparison between the performance parameters of the control system proposed in this paper and the existing control system in this field of technology is shown in Table 3.

**Table 3.** Comparison of the performance parameters of each control system.

| With/Without Feedforward Inputs of the Intelligent Algorithm | Speed Measuring Method | $V_{PP}$ | $V_{RMS}$ | Sampling Error | Accuracy |
|---|---|---|---|---|---|
| With | T method | 0.047 | 0.0030 | 0.001 | 0.0003 |
| | T method | 0.083 | 0.0093 | 0.001 | 0.0009 |
| Without | M method | 0.088 | 0.0140 | 0.056 | 0.0013 |
| | T method | None [1] | 0.0102 | 0.000025 | None |

[1] None means that the parameter was not mentioned in the study.

## 4. Discussion

The moving mirror is very important for the FTS, and the stability of its motion directly determines the performance of the FTS. When acquiring interferograms with a constant optical path difference sampling in the uniform speed zone, if the speed of the moving mirror fluctuates, the sampling error of the interferograms will be introduced, which will cause ghost lines to affect the spectral quality during spectral inversion. The speed fluctuations also introduce modulation noise into subsequent signal processing, which will also affect the spectral quality. Therefore, we need to strictly control the speed fluctuations of the moving mirror. The $V_{PP}$ and $V_{RMS}$ are two parameters that can intuitively reflect the speed fluctuations. The $V_{PP}$ represents the amplitude of instantaneous speed fluctuations, and the $V_{RMS}$ represents the effective value of the speed fluctuations. The better the

performance of the moving mirror speed control system, the smaller the $V_{PP}$ and $V_{RMS}$, the better the velocity uniformity, and the higher the spectral quality. From the experimental results, it can be seen that the moving mirror control system based on the feedforward inputs obtained by the intelligent algorithm significantly reduces the $V_{PP}$ and the $V_{RMS}$ and achieves a superior control effect, which is an effective technical solution to improve the performance of the interferometer.

## 5. Conclusions

Feedforward control is usually a control means to improve the performance of the control system, and generally, the feedforward inputs are obtained through a theoretical calculation based on the mathematical model of the controlled object. In the FTS of this paper, it is difficult to obtain accurate feedforward inputs through calculation. Therefore, an intelligent algorithm to acquire the accurate feedforward inputs is proposed, which can automatically and intelligently obtain the feedforward inputs based on the expected motion law. The feedforward inputs obtained by the intelligent algorithm are related to the real controlled object that is produced, while the controlled object already contains all the errors in the production process. Therefore, the feedforward inputs obtained by the intelligent algorithm combined with the speed feedback control can control the moving mirror more accurately and effectively, which means that the moving mirror system of the FTS in this paper achieves a better performance. Compared with the moving mirror system without feedforward inputs, the moving mirror control system based on the feedforward inputs obtained by the intelligent algorithm reduces the $V_{PP}$ by 43.4% and the $V_{RMS}$ by 67.7%, greatly improving the velocity uniformity of the moving mirror. Therefore, the control system based on the feedforward inputs obtained by the intelligent algorithm is a feasible and effective moving mirror speed control scheme for the FTS.

**Author Contributions:** Conceptualization, J.H.; methodology, J.D.; software, Y.H.; validation, Y.H., and Q.G.; investigation, Y.H.; resources, Z.W.; writing—original draft preparation, Y.H.; writing—review and editing, J.D.; supervision, J.D.; project administration, Z.W.; funding acquisition, Z.W. All authors have read and agreed to the published version of the manuscript.

**Funding:** This research was funded by China's National Key Special Earth Observation and Navigation Project "Atmospheric radiation hyper-spectral detection technology", Ministry of Science and Technology of the People's Republic of China (MOST), grant number YFB0500600 and Mid-infrared Observation System for Accurate Measurement of Solar Magnetic Field, National Natural Science Foundation of China, grant number 11427901.

**Data Availability Statement:** Data are contained within the article.

**Conflicts of Interest:** The authors declare no conflict of interest. The funders had no role in the design of the study; in the collection, analyses, or interpretation of data; in the writing of the manuscript; or in the decision to publish the results.

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
