# Peer review of "Based on the Feedforward Inputs Obtained by the Intelligent Algorithm the Moving Mirror Control System of the Fourier Transform Spectrometer"

_electronics, doi:10.3390/electronics12224568_

Round 1

Reviewer 1 Report (Previous Reviewer 2)

Comments and Suggestions for Authors

In the first submission, the authors addressed my comments accordingly. In this new version, I have no more comments; the article has the quality to be considered for publication. 

Author Response

Dear Reviewer,

Thank you very much for your recognition of my manuscript.

Best Regards

Reviewer 2 Report (Previous Reviewer 3)

Comments and Suggestions for Authors

Congratulations. I would like to recommend to accept current version. 

Author Response

Dear Reviewer,

Thank you very much for your recognition of my manuscript.

Best regards

Reviewer 3 Report (Previous Reviewer 4)

Comments and Suggestions for Authors

Dear authors, these are my comments about this work:

It is indicated that an intelligent algorithm is developed to obtain the feedforward inputs automatically but it is not clearly demonstrated how this process will be carried out in the development of this work.

A comparative table must be shown of the contributions that this algorithm has with respect to the state of the art; there are various performance metrics such as speed, sampling error, and accuracy to evaluate the control system, this must be reflected in the table with respect to to related works in the state of the art.

Figure 1 has low resolution, all figures must have the same format, the same happens with Figure 6.

There is no adequate discussion of the Vrms obtained in Figures 8 and 9, what impact do these results have?

We show equations in the discussion and conclusions section.

Best Regards

Author Response

Dear Reviewer,

Thank you very much for your positive and constructive comments and suggestions on my manuscript. Please see the attachment for my responses.

Best Regards

Round 2

Reviewer 3 Report (Previous Reviewer 4)

Comments and Suggestions for Authors

Dear authors,

I have received the new version of the document, I see it substantially improved, regarding the comparative table of the state of the art, consider that it must be related to existing works in the literature, where the metrics of execution time, sampling error and precision, and additional ones are visible.

 Best Regards

Author Response

Dear Reviewer,

Thank you very much for your comments and suggestions. Plaese see the attachment for details.

Best Regards

This manuscript is a resubmission of an earlier submission. The following is a list of the peer review reports and author responses from that submission.

Round 1

Reviewer 1 Report

Comments and Suggestions for Authors

While I appreciate the experimental aspect, presentation must be improved and overall results should be presented in a better form. But the main weakness lies in the fact that feedforward control in linear motors has been around for almost four decades, applying it to a specific problem is, in my opinion, not enough for a research paper to be published in a journal.

Comments on the Quality of English Language

English editing is needed, several sentences are not that much clear.

Author Response

Dear Reviewer,

Thank you very much for your positive and constructive comments and suggestions on my manuscript. The details of my responses are in the attachment. Please see the attachment.

Kind regards

Reviewer 2 Report

Comments and Suggestions for Authors

The present work addresses the design of a feedforward input plus a feedback controller for a speed control system. After the review, the following comments arise:

1.- The introduction needs to explain the feedback controllers commonly used to regulate servo control systems. More references should be included about it.

2.- It is not clear the main contribution. This must be clearly explained at the end of the introduction. The feedforward input is not a novel contribution.

3.- Why is the inductance L not included in Eq. (1)? Figure 2 includes the inductance.

4.- The stability and the robustness against disturbance of the Intelligent Algorithm must be proved.

5.- A model validation should be included to demonstrate that the feedforward input is correct for the real system.

6.- The Intelligent feedback controller's performance should be compared, at least, with the commonly used controllers' performance in servo control systems.

7.- The equations' grammar should be reviewed.

Comments on the Quality of English Language

The equations' grammar needs review.

Author Response

(The authors gave the same response as above.)

Reviewer 3 Report

Comments and Suggestions for Authors

The author presents the feedforward inputs for speed control system for FTS system. It is very import study for FTS performance improvements. I would like to encourage author making better description for this work.

1. On line 31, please define 'a little better'. I would like to ask to list specified result here. 

2. Line 86 na 87, replace Kf with K. Same comments for Kx, Kbe, etc.

3. Line 88, please full spell EMF.

4. Is the Kbe a constant number? If not, I recommend adding a short description about that.

5. In figure 3, does the clock period must be an integral multiple of the laser period?

6. The resolution of figure 5 is poor, please replot that.

7. The Y axis ranges in figure 5b, 5c, 5d are not suitable to evaluate the speed difference. Please narrow the Y range. Those are essential results in your testing.

Author Response

(The authors gave the same response as above.)

Reviewer 4 Report

Comments and Suggestions for Authors

Dear authors,

These are my comments regargind this work. 

It is not shown that the feedforward inputs obtained by the intelligent algorithm are more suitable for the real object controlled in the results stage.

The introduction lacks a survey of the state of the art of related or comparative works based on FTS.

The speed control law is not properly described in the development section.

The developed system of equations is not adequately described, it tries to explain it with an algorithm flow for the acquisition of the feedforward control quantity, but it is extensive and difficult to understand.

I believe that more depth is required in the development stage to justify the contribution of this work.

Best Regards

Author Response

Dear Reviewer,

Thank you very much for your positive and constructive comments and suggestions on my manuscript. The details of my responses are in the attachment. Please see the attachment.

Best regards

Round 2

Reviewer 1 Report

Comments and Suggestions for Authors

Dear authors, I appreciated the changes you introduced, in particular the expansion of use case explanation in lines 66-82.

I think however that the only somehow noticeable improvement lies in peak-to-peak velocity, while the other figures are substantially negligible.

I see a reference to "collected data" (line 297), which seems to be quite interesting would you please expand it?

Comments on the Quality of English Language

Starting from the title itself there I think there an extensive editing is needed.

Author Response

Dear Reviewer,

Thank you very much for your positive and constructive comments and suggestions on my manuscript. The details of my response are in the attachment. 

Kind Regards

Reviewer 2 Report

Comments and Suggestions for Authors

After this second review, the authors have addressed the comments made in the first review, but I have two more comments:

1.- The article's content should be summarized at the end of the introduction.

2.- As Reviewer 4 suggested, the design of the algorithm should be clarified for readers. The authors answered the reviewer's comment but did not add a modification or improvement in the algorithm explanation.

Author Response

(The authors gave the same response as above.)

Reviewer 4 Report

Comments and Suggestions for Authors

 Greetings,

I have thoroughly examined the revised version of the document. However, I must point out some areas that require improvement in order to enhance its academic quality. Firstly, the Introduction lacks a comprehensive comparison of metrics with existing works in the field. Additionally, I have observed that only one reference (reference 28) has been added to support the discussion.

The inclusion of Figure I, which represents the speed control system, is intriguing. However, the stages developed in this work are not adequately described. Consequently, it is challenging to grasp the significance and intricacies of the system.

Furthermore, Figure 3 does not appear to contribute significantly, as it exhibits a nearly constant behavior. Thus, its inclusion in the document seems unnecessary and fails to add substantial value to the overall discussion.

Moreover, I believe that the equations used for the Feedforward control in this contribution lacks adequate description. The precise manner in which they are employed remains unclear, hindering the understanding of their role in this work.

I suggest a greater effort in accurately describing the contribution of this study, as well as providing more comprehensive explanations for each of the sections.

Best Regards,

Author Response

Dear Reviewer,

Thank you very much for your positive and constructive comments and suggestions on my manuscript. The details of my response are in the attachment. 

Best Regards

Round 3

Reviewer 1 Report

Comments and Suggestions for Authors

Overall I still don't see the article fitting a particular target: in terms of control systems the result is pretty trivial, while the specific application is not valorized properly because it lacks a detailed explanation of the experimental setup and techniques, probably focusing on experimental results would be the best way to "pack" this paper.

Comments on the Quality of English Language

There's still room for increasing readability,

Reviewer 4 Report

Comments and Suggestions for Authors

Dear authors,

I have reviewed the new version, I see that some paragraphs have been rewritten in a better way, the main concern lies in the information provided by the figures, most of them on the X axis are not well centered and the information that is graphed does not show a contribution overwhelming.

 Another factor is the systems of equations that, when reading the document, are not widely related to what it is trying to describe, nor are the systems of equations related to the figures shown in the work. I believe that a fundamental restructuring is needed to clearly reflect that the performance tests carried out on Feedforward control are a contribution on this subject.

Regards.